# Adaptive Radiotherapy: Next-Generation Radiotherapy

**DOI:** 10.3390/cancers16061206

**Published:** 2024-03-19

**Authors:** Olga Maria Dona Lemus, Minsong Cao, Bin Cai, Michael Cummings, Dandan Zheng

**Affiliations:** 1Department of Radiation Oncology, University of Rochester, Rochester, NY 14642, USA; olgamaria_donalemus@urmc.rochester.edu (O.M.D.L.); michael_cummings@urmc.rochester.edu (M.C.); 2Department of Radiation Oncology, University of California Los Angeles, Los Angeles, CA 90095, USA; minsongcao@mednet.ucla.edu; 3Department of Radiation Oncology, University of Texas Southwestern Medical Center, Dallas, TX 75390, USA; bin.cai@utsouthwestern.edu

**Keywords:** adaptive radiotherapy, adaptive replanning, treatment adaptation, MRgRT, CBCT, PET, IGRT, personalized medicine

## Abstract

**Simple Summary:**

Radiotherapy, a crucial cancer treatment, has evolved significantly over the years. Traditionally, treatment plans were based on initial scans used throughout the treatment course, accounting for changes in the patient’s anatomy by additional margins to targets. However, the field has moved towards decreasing margins with the advancement of delivery and targeting accuracy in order to decrease toxicity, and the increasing use of image guidance has illuminated patient anatomical changes such as organ deformation, weight loss, tumor shrinkage, and even biological changes that are unaccounted for by the conventional approach. Adaptive radiotherapy (ART) addresses this by adjusting treatment plans according to these changes. ART can be conducted in two ways: online (adjustments made during treatment sessions) and offline (adjustments made between treatment sessions). Advances in technology, especially in medical imaging (CT, MRI, and PET scans) and artificial intelligence, have made ART more feasible and efficient. ART offers more precise cancer treatment by adapting to changes in the patient’s body, leading to better outcomes with fewer side effects.

**Abstract:**

Radiotherapy, a crucial technique in cancer therapy, has traditionally relied on the premise of largely unchanging patient anatomy during the treatment course and encompassing uncertainties by target margins. This review introduces adaptive radiotherapy (ART), a notable innovation that addresses anatomy changes and optimizes the therapeutic ratio. ART utilizes advanced imaging techniques such as CT, MRI, and PET to modify the treatment plan based on observed anatomical changes and even biological changes during the course of treatment. The narrative review provides a comprehensive guide on ART for healthcare professionals and trainees in radiation oncology and anyone else interested in the topic. The incorporation of artificial intelligence in ART has played a crucial role in improving effectiveness, particularly in contour segmentation, treatment planning, and quality assurance. This has expedited the process to render online ART feasible, lowered the burden for radiation oncology practitioners, and enhanced the precision of dynamically personalized treatment. Current technical and clinical progress on ART is discussed in this review, highlighting the ongoing development of imaging technologies and AI and emphasizing their contribution to enhancing the applicability and effectiveness of ART.

## 1. Introduction

Radiotherapy is a major modality for treating cancer, used in more than 50% of all cancer patients either alone or in conjunction with other modalities [1]. Radiotherapy is an early form of personalized medicine that involves prescribing treatments specific to different diseases and stages and developing treatment plans for individual patients. These treatments are based on evidence-based standards and are individually optimized for each patient, taking into account their anatomy. The goal is to deliver the most effective dose to kill cancer cells or impede their proliferation while minimizing harm to nearby healthy tissues and organs at risk (OARs).

The process of radiotherapy treatment planning involves obtaining the patient’s anatomy through medical imaging during treatment simulation and then designing the customized beam arrangement, beam number, geometry, intensity, and modulation during plan optimization. This allows for a personalized approach to treatment. Currently, computed tomography (CT) is the primary method used in radiation for treatment simulation. Sometimes, magnetic resonance imaging (MRI) or positron emission tomography (PET) are also used in addition to CT.

In the last decades, improvements in technology have improved the planning and delivery of radiation, making it more precise and accurate [2,3]. Nevertheless, the conventional paradigm has failed to consider a crucial aspect: the assumption that the patient’s anatomy remains largely constant during the course of the treatment or that any discrepancy can be accounted for by the planning margins. In reality, significant anatomical changes can occur due to daily variations, such as changes in organ fillings in the abdominal and pelvic regions that lead to relative positional change and deformation, as well as broader trends like fluctuations in patient weight and changes in tumor volume and shape [4,5,6,7]. Adaptive radiotherapy (ART) has therefore been suggested as a solution to adjust treatment plans according to anatomical changes and ensure the planned optimal therapeutic ratio is maintained throughout the treatment course [8]. While the concept is not new, it was not until the recent few years, with the advent of systems capable of executing online ART with integrated systems that support all the required steps and the help of artificial intelligence (AI), that ART started to gain popularity and show the promise of ushering radiotherapy into a new paradigm. In this narrative review, we aim to provide a comprehensive guide on ART for healthcare professionals and trainees in radiation oncology and those from other oncology fields. With a brief discussion of the evolution of radiotherapy, we will emphasize the significance of ART in accommodating patient anatomical changes during treatment and discuss the motivations for ART. Different types of ART, including offline and online methods, and their respective workflows, technologies, and clinical applications will be discussed. Highlighting the role of AI in enhancing ART and improving patient outcomes, we will discuss in detail all three major ART technologies currently available, based on MRI, cone-beam CT (CBCT), and PET. This review will also present a thorough summary and discussion on the implications of ART in treating various cancers, including cervical, lung, prostate, and head and neck cancers, demonstrating ART’s efficacy in optimizing treatment accuracy and minimizing risks.

## 2. Evolution of Radiotherapy and Why We Need ART

Over the past many decades, radiotherapy has evolved in sophistication to achieve increasingly improved accuracy and precision. With these advancements, handling daily anatomical changes through ART became necessary owing to the capability to conformally and accurately target the tumor and the ever-shrinking margins [9,10].

The anatomical changes inevitably increase the difficulties in delivering precise treatment. This has especially become a specific concern in the current era of sophisticated radiotherapy, when doses are precisely tailored to the tumor while minimizing radiation exposure to nearby organs at risk. Figure 1 demonstrates the notion by showing in A1 and B1 how the anatomy on the day of treatment may differ from the anatomy during the simulation and how this can affect the delivery of radiation doses in different radiotherapy techniques, ranging from two-dimensional treatments to ART. Panels A2 and B2 provide a straightforward example of the 2D treatment technique, in which a sizable, non-conforming region (depicted in yellow) is exposed to radiation. While the extensive irradiated region ensures that the treatment plan can handle changes in anatomy, it also restricts the amount of radiation that can be administered to the tumor due to the excessive dose and potential harm to the neighboring healthy tissues and OARs. Panels A3 and B3 illustrate the application of more sophisticated methods, such as 3D CRT, which involves shaping the irradiated region (depicted in yellow) to match the tumor’s shape precisely. This approach results in a smaller irradiated area surrounding the tumor, hence minimizing the harmful effects on nearby healthy tissues and OARs. Panels A4 and B4 illustrate the application of highly sophisticated approaches, namely intensity-modulated radiotherapy (IMRT) and volumetric-modulated arc therapy (VMAT) with image-guided radiotherapy (IGRT). These techniques enable the irradiated area (depicted in yellow) to conform even more precisely to the tumor, while simultaneously minimizing radiation exposure to neighboring organs at risk (OARs). By utilizing advanced contemporary methods, it is possible to administer a substantial dose that effectively kills tumors without compromising nearby OARs or causing significant adverse effects. Nevertheless, as demonstrated in panels A3-4 and B3-4, when there is a disparity between the anatomy on the treatment day and the anatomy during the simulation, as the treatment plan becomes increasingly sophisticated and tightly conformal to the original targets, the anatomical changes can lead to a greater deviation from the initial intention. Consequently, the coverage of the tumor dose is compromised, and the OARs receive a higher dose than planned, thereby reducing the therapeutic ratio. Panels A5 and B5 illustrate an instance of ART in which the treatment plan is readjusted according to the anatomy of the patient on the day of treatment. This ensures that the tumor receives a highly precise dose while minimizing radiation exposure to nearby OARs. ART signifies a fundamental change from conventional radiotherapy, as it involves dynamically reoptimizing the treatment plan to accommodate changes in the patient’s anatomy. This guarantees the preservation of accurate radiation dosage to the tumor while specifically protecting nearby OARs, thus optimizing the therapeutic ratio that can be achieved with advanced radiotherapy technology.

## 3. Frequency, General Workflow, and Offline vs. Online ART

The frequency of conducting ART can be adjusted based on the rate of anatomical changes. Situations with gradual changes may require less frequent ART, while situations with random variations may necessitate more frequent ART. ART can be conducted in two distinct manners—offline ART and online ART. In the context of offline ART, the images are obtained and the ART planning process is then carried out between treatment fractions, to be utilized in future sessions. Conversely, in the case of online ART, the ART workflow is performed within the same session or fraction as the images utilized for adaptation, specifically in relation to anatomy [11,12]. Table 1 summarizes a comparison between offline and online ART. While offline ART can be carried out using the same software and workflow used in conventional radiotherapy, with a process taking hours to days, online ART often uses a specialized software/hardware platform with automated steps and a streamlined workflow that reduces the process to several to several tens of minutes. Offline ART can address gradual anatomy changes during the treatment course, and online ART can address more random and abrupt interfractional anatomy changes. However, as even online ART does not have an instantaneous adaptive workflow but takes some time to complete, there could still exist instantaneous intrafractional anatomy changes that are not fully addressed by the adaptation, such as the continuing bladder filling during a treatment session and the movement or deformation of abdominal gas pockets and organs. Ideally, real-time ART that provides plan adaptation based on real-time anatomy would fully address both interfractional and intrafractional anatomy changes. However, such technology does not currently exist.

Figure 2 illustrates a comprehensive workflow of ART. Just like the original treatment planning procedure, it begins with obtaining images of the new anatomy. In addition to simulation imaging used for the original treatment planning, other in-treatment-room imaging modalities can also be utilized, such as CBCT, in-room CT, MRI, and PET. Contours must be developed for the targets and OARs using fresh anatomical images. By utilizing the updated anatomy and outlines, it is possible to reconstruct the delivered dose according to the initial treatment plan. A decision on whether to adapt is subsequently taken after analyzing the reconstructed dose on the new anatomy. When adjusting, a fresh treatment plan is created based on the altered anatomy. Occasionally, particularly in the case of online ART, the production of a new treatment plan may occur simultaneously with the reconstruction of the previous plan’s dosage in order to streamline the workflow. Regardless of the scenario, it is necessary to perform plan quality assurance (QA) before administering the new plan.

ART incorporates the changes in anatomy and formulates adaptive treatment plans. Finally, an extra supplementary phase in the ART workflow is dose accumulation. In the process of dose accumulation, the cumulative dose is calculated by taking into account both the anatomical changes and modifications in the treatment plan, in order to construct a precise representation of the administered dose.

### 3.1. Offline ART

Offline ART could utilize sophisticated imaging modalities such as CBCT, MRI, or PET imaging to acquire high-resolution images of the patient’s anatomy [13,14]. But it could also rely on a re-simulation using the simulation CT as in conventional radiotherapy. These images are utilized to evaluate the changes in tumor dimensions, shapes, and location over the course of therapy and those of the OARs and the body. Offline ART monitoring systems can employ rigid registration, deformable image registration (DIR), contour propagation, and dose accumulation techniques to enable radiation oncologists to efficiently evaluate contours or treatment plans using periodically collected imaging data. If changes are observed that indicate a loss of tumor dose coverage or violation of OAR tolerances, a request is made for a new treatment plan. In certain instances, it is possible to foresee and plan for changes, such as in the case of bladder malignancies, where varying degrees of bladder filling can be predicted. Treatment plans can be made for a few bladder sizes, and the treatment plan for a particular day can be adapted by selecting the plan that most closely matches the bladder filling at the time of treatment.

Offline ART typically entails extensive collaboration among radiation oncologists, medical physicists, dosimetrists, and radiation therapists to effectively devise and execute adaptive therapy strategies. QA processes are implemented to validate the precision of the procedures, guaranteeing that the therapy is delivered with exactitude according to the plan. This encompasses routine inspections of imaging equipment, software for monitoring treatment, and systems for delivering treatment. Offline ART enables highly personalized treatment, taking into account the precise alterations in a patient’s anatomy and tumor attributes. This strategy reduces the likelihood of administering insufficient doses to the tumor or excessive doses to healthy organs and is commonly employed in situations when anatomical variations are anticipated but not occurring daily. It is frequently used in the treatment of head and neck cancers, where many clinicians would use offline ART to reoptimize the planned dose to accommodate a patient’s weight loss and/or tumor shrinkage. Some perform this once during the whole treatment course, while others may adapt biweekly or weekly.

### 3.2. Online ART

Online ART also relies on on-couch volumetric imaging techniques, such as CBCT, MRI, and PET imaging, which allow for frequent and immediate visualization of the patient’s anatomy and tumor position before and during each treatment session [10,15]. State-of-the-art online ART systems utilize advanced treatment planning software that can quickly process imaging data and make in-session adjustments to the treatment plan. These software tools are designed to optimize the radiation beam placement and intensity to accommodate daily variations in the patient’s anatomy. AI and machine learning algorithms are increasingly integrated into online ART platforms to accelerate tumor and OAR segmentation and to increase the efficiency of in-session adaptations. Intrafractional or real-time monitoring is available in some online ART platforms which involves the continuous tracking of the patient’s anatomy and tumor position during radiation delivery. This allows for immediate corrections if there is any significant deviation from the treatment plan. In terms of clinical application, online ART is employed in clinical scenarios where the tumor’s location and shape may vary on a day-to-day basis, such as lung, prostate, and liver cancers, as well as situations where the proximity to critical structures necessitates precise targeting. QA measures are in place to ensure the accuracy and safety of online ART procedures, often including an independent calculation system linked to the treatment unit and immediate analysis of the log files that contain beam delivery information. All these checks are conducted in addition to routine checks of equipment and software used in online adaptation. Online ART also requires close collaboration among radiation oncologists, medical physicists, dosimetrists, radiation therapists, and other healthcare professionals to optimize the treatment and provide in-session adjustments. This modality allows for the most patient-specific treatment by adapting to anatomical changes that occur at every session. This minimizes the risk of under-dosage of the tumor and minimizes radiation exposure to healthy tissues.

### 3.3. Resource Considerations and Role of AI

The implementation of ART has the potential to significantly increase the burden for specialists in the field of radiation oncology. If a course of prostate radiation consisting of 40 daily fractions were to be treated using daily online ART, the burden for treatment planning and QA would rise by a factor of 40 compared to the standard approach. Furthermore, the considerable duration, ranging from hours to days, required for conventional radiotherapy treatment planning, coupled with the labor-intensive tasks performed by doctors and planners, renders online ART unfeasible. Fortunately, the implementation of automation using diverse AI techniques has effectively resolved these concerns. The utilization of AI-based auto-segmentation and DIR has facilitated the process of target and OAR recontouring on the session anatomy, making it more efficient, rapid, and reasonably accurate [16,17]. AI techniques have been employed to produce synthetic CT images using CBCT or MRI scans for the purpose of treatment planning and radiation dose computation [17,18]. AI-driven automated treatment planning has expedited the process of reoptimizing new treatment plans, reducing the time required to a matter of minutes [19]. AI-driven dose calculation and error detection have facilitated the implementation of calculation-based patient QA, eliminating the necessity for patients to undergo phantom-based QA while on the treatment table [20]. In addition, radiomics and AI-based assessment tools integrate the patient’s imaging and other test results throughout the treatment process to monitor and predict treatment toxicity and outcomes. This allows for personalized assessment and adaptation decisions based on ongoing evaluation [21,22].

## 4. Three Major Imaging Modalities for Online ART

At present, online ART technologies based on three major imaging modalities are accessible for commercial use, as seen in Figure 3. The MR-based platform (Figure 3A) was the initial option, now offered by several manufacturers and models. It was followed by the CBCT-based platform (Figure 3B) and most recently by the PET-based platform (Figure 3C). Currently, there is only one vendor/model available for either the CBCT-based or the PET-based system. However, the PET-based system is still awaiting FDA approval for online ART applications.

Although the treatment planning software of these platforms does not currently allow it, these systems have the potential to offer real-time adaptability. For example, the PET-guided system may in the future use obtained PET-CT images to make real-time adjustments to the target volume and OARs. After the redefinition of outlines and the analysis of PET signals, the plan adaption may be carried out in real time with further technology advancements.

Table 2 summarizes a brief comparison of the three online ART technologies. More will be discussed about the technical and clinical aspects of these technologies in the following sections in terms of workflow, strengths and limitations, practical considerations for implementation and QA, challenges and uncertainties, etc.

### 4.1. MRI-Based Online ART

MRI’s exceptional ability to distinguish soft tissues allows for the accurate evaluation of anatomical changes, making it an optimal tool for online adaptive replanning in radiotherapy. As shown in an example case in Figure 4, MR-LINAC provides good soft-tissue contrast, highlighting the daily changes in the relative position and shape of the tumor and OARs for a pancreatic cancer case, and plan adaptation helps maintain the optimal tumor coverage vs. OAR sparing.

Two commercial MR-LINAC systems provide the necessary capabilities for online MR-guided planning adaptation. While the technological implementation and specifications of these systems may differ, their online adaptation mostly depends on anatomic MRI guidance. A legacy system was ViewRay MRIdian (ViewRay, Mountain View, CA, USA) integrating a 0.35 T MRI with a 6MV FFF LINAC, replacing their Co-60-based unit originally introduced in 2014. As the earliest available MR-guided radiotherapy systems, they stimulated numerous investigations and contributed to much of the MRI-based ART experience. Currently available in the market is the Unity system, with a 1.5 T MRI and a 7MV FFF LINAC by Elekta (Elekta, AB, Stockholm, Sweden), which was the first commercial MR-LINAC released in 2018 shortly before MRIdian.

**Figure 4 cancers-16-01206-f004:**
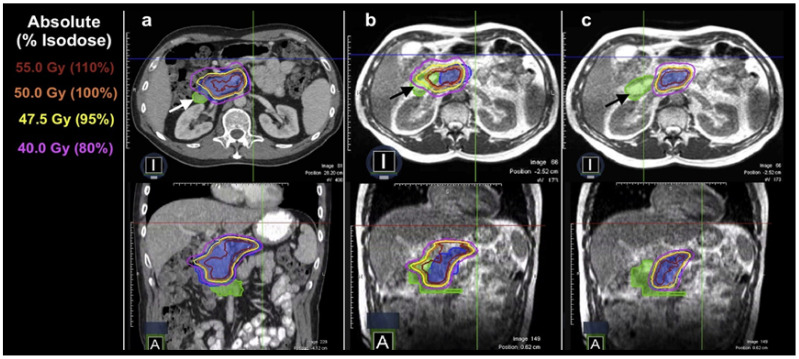
(**a**) Initial plan met all organ-at-risk constraints for a patient with a pancreatic tumor (blue color wash) based on the anatomy from the initial CT simulation. (**b**) Application of the plan to the daily MRI set resulted in a violation of hard duodenal (green color wash) constraints. (**c**) Daily adaptive planning achieved the resolution of the OAR constraint violation to the duodenum (marked with arrows) while preserving target volume coverage. Reprinted/adapted with permission from “Simulated Online Adaptive Magnetic Resonance–Guided Stereotactic Body Radiation Therapy for the Treatment of Oligometastatic Disease of the Abdomen and Central Thorax: Characterization of Potential Advantages” by Henke et al. 2016, International Journal of Radiation Oncology* Biology* Physics, 96(5), Copyright 2016 by Elsevier [23].

The adaptive workflows on the 1.5 T MR-LINAC system Unity may be classified into two fundamental strategies: adapt to position (ATP) and adapt to shape (ATS) [11]. The ATP process is mostly a virtual method to circumvent the limitation of couch movements during a couch shift. The isocenter shift is determined by performing rigid registration of daily MR images with the reference images. Subsequently, the initial treatment plan is adjusted to achieve the desired target coverage by modifying the shape and/or weighting of the MLC segments. The ATS workflow involves a thorough process of reoptimization, taking into account the actual anatomy and adjusted shapes. However, the ATS procedure does not include the process of reconstructing the dosage on the new anatomy. Instead, it directly performs dose reoptimization after making contour alterations. The selection of either ATP or ATS workflows is contingent upon the daily anatomical evaluation or institutional protocols and can be integrated in particular clinical situations. For the MRIdian system, following the daily acquisition of MR images, a conventional image-guided patient setup can be conducted on the 0.35 T MR-LINAC system. The initial treatment plan can be reassessed using the daily MR images and adjusted contours to determine if the original plan is still suitable for the current daily anatomy. If a dosimetric assessment, using the projected dose, determines that adaptive planning is required, the initial treatment plan will be reoptimized. To gain a comprehensive understanding of the workflow on various platforms, one might refer to the elaborate explanations provided in ICRU report No. 97 [24]. The successful integration of online MR-guided ART planning in clinical practice requires careful process management and the construction of a robust QA program. This entails tackling overarching aspects for online adaptive planning, encompassing automated and robust planning techniques as well as patient-specific QA approaches. Furthermore, it is necessary to develop specialized strategies that are customized for the utilization of MRI as the principal imaging modality for planning purposes. These tasks involve guaranteeing the quality of MR images and properly measuring electron density to calculate doses with precision.

Online ART planning involves all the necessary phases of traditional treatment planning but is carried out quickly while the patient is undergoing the procedure, requiring a highly efficient and reliable execution. In order to make contouring more efficient, DIR is often used to transfer contours from reference to daily MR images. Nevertheless, due to the inherent uncertainties in DIR, it is essential to conduct visual assessments and make necessary manual modifications. Another method is improving a specific group of structures near the objective, reducing the need for manual intervention [25]. Furthermore, established criteria can be used to construct derived shapes, such as target expansions or optimization tuning structures. This process improves efficiency and minimizes human operational errors when conducting online planning. Multiple clinical investigations [26] have identified contour segmentation as a significant obstacle in the online ART procedure, despite the implementation of various techniques. Recent studies on AI-based auto-segmentation demonstrate potential in greatly accelerating the process of segmentation. Nevertheless, the incorporation of these methods into clinical practice faces numerous obstacles, such as concerns regarding the accuracy of training data, inconsistencies across different institutions and observers, and logistical complexities in interfacing with treatment planning systems.

Efficient and robust methods for optimizing and calculating doses are crucial for quickly adapting plans. The beam angles and optimization parameters from the initial plan can serve as the starting point for reoptimization. However, this necessitates the initial planning to be robust enough to handle significant daily fluctuations. If the weight assigned to an OAR that is located far from the target volume is not enough during simulation, it may lead to inferior quality of the treatment plan as the OAR comes closer to the target during the adaptive fraction. In order to tackle possible planning situations that may not be optimal, researchers have suggested solutions such as grouping several OARs or merging them into a single optimization structure depending on their closeness to the planning target volume (PTV) [25].

Conventional pretreatment measurement-based patient-specific QA is not feasible in the online adaptive workflow since the patient must stay in the treatment posture. Current patient-specific QA solutions mostly depend on secondary absorbed dose calculations utilizing simplified calculation tools offered by manufacturers or third-party software. It is advisable to conduct initial measurements on a representative group of patients in order to establish a connection between measurement- and calculation-based methods. This practice helps to increase confidence in the accuracy of secondary absorbed dose calculations [12]. Real-time treatment delivery monitoring has been investigated as a substitute for pretreatment QA. Analysis conducted using machine log files on the 1.5 T MR-LINAC system has demonstrated a significant association with measurement-based QA, providing a high level of sensitivity in detecting errors [27]. The integration of machine log data and cine images has the potential to enable the reconstruction of the absorbed dose that was provided, while taking into consideration any motion that occurred during the treatment [28].

Online MRI-based adaptive planning is primarily a system that relies solely on MRI scans. This is because it is not feasible to obtain a CT scan while the patient is in the treatment position just before treatment. Methods for acquiring electron density data encompass bulk density assignment, deformable registration using the simulation CT, and voxel-intensity-based synthetic CT. Utilizing deformable registration to align the simulation CT with the MRI provides a direct method for acquiring patient-specific electron density information at the early planning stage. This is especially useful when the simulation CT and MRI are obtained within a short period of time with few anatomical alterations. Nevertheless, when it comes to online adaptive planning, the process of combining daily MR with simulation CT might potentially result in inaccuracies in electron density mapping, particularly when there are anatomical alterations such as differences in organ filling and the presence of air cavities inside the gastrointestinal tract. Manual examination, modifications, and density overrides may be required. An alternate method entails creating synthetic CT scans directly from the daily MR images using voxel-based inference. This approach offers the benefit of a strong anatomical correlation, which has the potential to improve the precision of electron density mapping and dose calculation [29].

The effectiveness of ART depends on the acquisition of high-quality daily images to identify anatomical alterations and ensure the accurate delineation of tumors and organs. Due to the vulnerability of MR images to artifacts and spatial distortion, it is crucial to regularly perform QA on the imaging system in the room. It is strongly advised to regularly assess image performance, including geometric distortion, image quality, and artifacts, in order to promptly detect any potential imaging problems. Implementing automated image QA analysis can accelerate the QA process, offering a fast and reliable method for regular tests to identify and resolve any errors, hence guaranteeing optimal performance for online AI systems.

Currently, the primary clinical application of adaptive planning on the MR-LINAC system is based on anatomical variances. Nevertheless, there is potential in the field of biologically guided ART. This approach is convincing because alterations at physiological and molecular levels clearly define the innate biological reaction to radiation therapy, providing potential predictive significance and the ability to adjust treatment regimens. Recent research suggests that it is possible to quantify Quantitative Imaging Biomarkers (QIBs) using MR-LINAC systems, even in situations with weak magnetic fields [30]. However, a major obstacle exists in developing a thorough QA program to guarantee accurate measurements with strong consistency and replicability. Another crucial challenge involves formulating strategies and techniques to smoothly incorporate biological data into the decision-making procedures of the adaptive workflow, defining the optimal timing and approach for adaptation. There is a requirement to provide tools for efficient online decision-making by means of visual interpretation and evaluation. Furthermore, it is imperative to design mechanisms for the efficient incorporation of biological information with anatomical adaptation. These developments are crucial for fully fulfilling the potential of biologically guided ART in clinical practice.

### 4.2. CBCT-Based Online ART

The first CBCT-based ART system, Varian Ethos, was introduced in 2020 (Varian Medical Systems, Palo Alto, CA, USA). This system uses AI for contour segmentation, a machine learning-enhanced treatment planning system, an intelligent optimization engine (IOE), for adaptive plan generation, and improved CBCT image quality to create online adaptive plans within a typical timeframe of 15–25 min [31].

The CBCT-guided ART method starts similarly to any non-adaptive treatment by obtaining a planning CT scan and developing a treatment plan, referred to as the “reference plan”. At each treatment session, a CBCT image is obtained and subsequently segmented into distinct organs and skeletal structures using automated segmentation. The system’s pre-trained deep learning auto-segmentation model generates structures, referred to as “influencers”, which are either close to or within the target area. Within the adaptive workflow, the user has the ability to evaluate and modify the influencers. These influencers have an impact on the deformable creation of the targets, which may also be manually modified if necessary. The remaining structures are created by a DIR process that aligns the planned CT with the session CBCT. Using the outlined patient’s anatomy in the specified session, the system produces new dose distributions for OARs and targets to compare with the reference plan. The comparison involves two additional plans: the “scheduled plan”, which is the reference plan recalculated using the new anatomy, and the “adaptive plan”, which is a deliverable plan reoptimized on the new anatomy based on a prioritized list of clinical goals. Afterward, the radiation oncologist evaluates the two new plans against the reference plan and chooses one to be utilized. If the adaptive plan is selected, it undergoes a QA procedure, typically based on calculations, to ensure that the patient can remain on the table and the procedure can be performed in the same session. Ultimately, the selected treatment regimen is implemented on the patient, and this sequence is replicated for all following treatments with a dose accumulation constructed by the platform for an accurate delivered dose record [32].

Conventional CBCT, which is used to image patients for CBCT-based ART, has inferior image quality compared to fan-beam CT used for planning, mostly due to higher levels of scatter radiation. To overcome this limitation, the computation of the dose needs to be carried out on a synthetic image generated by applying deformable image registration to align the planned CT with the daily CBCT [33]. The development of iterative CBCT (iCBCT) has enhanced the overall quality of CBCT images, leading to more accurate CBCT-based IGRT. In addition, a forthcoming Ethos 2.0 combined with the new HyperSight CBCT technology (Varian Medical Systems, Palo Alto, CA, USA) anticipated for release in 2024 would possess innovative attributes including a very fast acquisition time down to 6 s, an expanded field of view of up to 70 cm, reduced artifacts, and advanced reconstruction procedures. As shown in the example images in Figure 5, HyperSight provides markedly improved image quality compared with conventional CBCT in terms of higher image contrast, reduced noise, and mitigated artifacts, which may arguably possess image quality almost comparable to the simulation CT. As such, in the new workflow, adaptive planning will be carried out directly on HyperSight CBCTs, bypassing the need for a synthetic CT. The utilization of these novel CBCT images for treatment planning and dose calculation will enable the resolution of some challenges associated with CBCT-based adaptive approaches.

The primary benefit of CBCT-based ART is its cost-effectiveness, which allows for widespread availability of these systems. Additionally, CBCT-based ART offers shorter treatment sessions compared to MRI or PET. Furthermore, it provides a comparably simpler and more seamless integration with emerging LINAC technologies and X-ray-based AI. The ability to directly plan and calculate dose using the images provided by the new HyperSight CBCT is particularly desirable and potentially eliminates the uncertainty associated with utilizing a synthetic CT. Conversely, CBCT is less effective at differentiating soft tissues that have identical X-ray attenuation coefficients, resulting in reduced contrast in soft-tissue structures. In regions characterized by intricate anatomy or in places where the accurate identification of soft-tissue structures is essential, the poor differentiation of soft tissues in CBCT images may present difficulties. On the other hand, the CBCT-based online ART system Ethos has been found to provide sufficient image contrast and accuracy for conducting effective online ART on abdominal, pelvic, head and neck, and other cancers [34,35,36,37,38]. Furthermore, CBCT lacks the capability to reveal details regarding metabolic activity or functional properties of tissues, as its main purpose is to visualize anatomical structures. Having information regarding metabolic activity could be beneficial for refining treatment plans in specific cancer treatment scenarios, particularly in cases involving ART where real-time adjustments are made depending on changes in tumor features.

To address these limitations, it may be advantageous to utilize a combination of imaging modalities during the treatment planning phase. CBCT can be utilized for primary positioning and localization objectives; however, alternative modalities such as MRI or PET may offer supplementary data for the precise identification of targets and evaluation of tumor attributes. Alternatively, radiomics and other AI approaches may also be used to leverage the correlation between biological information and CT images to explore CT-based biological guidance.

It is worth noting that imaging technologies and hybrid imaging systems are constantly evolving in order to overcome their inherent limitations and provide more comprehensive information for radiation therapy treatment planning. Medical professionals and scientists are continuously researching methods to combine various imaging techniques to improve the accuracy and efficiency of radiation treatments.

### 4.3. PET-Based ART

Both MR-guided and CBCT-guided ART have demonstrated encouraging outcomes, indicating potential advantages in terms of increasing the radiation dose to the target area and reducing radiation exposure to healthy tissues. At present, both MR-guided and CBCT-guided ART predominantly prioritize morphological characteristics, sometimes disregarding biological data. Combining functional and anatomical information through the use of PET (PET-CT) offers clear benefits over relying exclusively on anatomical imaging. PET, with its extensive track record in staging and evaluating therapy responses in different settings and types of tumors, offers useful insights in the field of oncology. PET has played a crucial role in radiation by aiding in target delineation, treatment planning, image-guided radiotherapy delivery, and functional modification for certain treatment sites. Anatomically adaptable radiation therapy focuses on modifying radiation doses according to changes in the structure of targets and/or OARs. In contrast, biologically adaptive radiation therapy aims to incorporate knowledge of tumor biology or functional aspects of OARs into the treatment and adaptive plans.

Traditionally, PET scanners have been used for biology-guided ART, which usually occurs in offline settings. The functional adaptation strategy for targeting involves identifying certain functional characteristics, such as tumor metabolism or hypoxia, and then altering or enhancing a specific area. On the other hand, the adaptation for sparing normal tissue relies on identifying functional alterations or damages in OARs. The recent advancements in PET technology, including the implementation of innovative detectors to enhance scanner performance, the use of AI to improve imaging reconstruction and enhancement, and the application of data-driven motion corrections, offer new possibilities in terms of detecting lesions, quantifying imaging results, and integrating with radiotherapy planning and treatment.

The PET-LINAC, RefleXion X1 (RefleXion Medical, Howard, CA, USA), represents a notable breakthrough in the field. It demonstrates the integration of a ring gantry LINAC and a PET-CT scanner in this groundbreaking technology, which is the first commercial implementation of PET-guided radiotherapy. The X1 system is a small radiation treatment system comprising a 6MV FFF LINAC installed on a rotating O-ring gantry that spins at a speed of 60 RPM. Additionally, there are two PET detector arrays positioned at 90-degree angles to the MV beamline. It facilitates the immediate identification and handling of molecular signals, directing the administration of radiation beams. Using this technology as a foundation, a novel therapeutic approach called SCINTIX therapy has been devised. A clinical trial conducted under an Investigational Device Exemption (IDE) has substantiated the safety and efficacy of SCINTIX therapy in the treatment of bone and lung lesions, leading to its approval by the FDA for these particular uses. Figure 6 illustrates a SCINTIX radiation plan that utilizes PET data to guide treatment for a patient with lung cancer. The optimal strategy considers both dosimetric limitations and the dispersion of PET signals. The bounded dose–volume histogram (DVH) accounts for uncertainty in both geometric and FDG intake. During the delivery process, the system continuously monitors and analyzes real-time signals, making necessary adjustments to the firing positions in order to obtain the intended dosage as per the initial design.

Undoubtedly, continuing progress in the hardware and software of radiotherapy technology will continue to improve the feasibility and simplicity of adopting PET-guided ART in clinical settings, both offline and online. This advancement is positioned to facilitate further clinical trials exploring the effectiveness of PET-guided ART.

## 5. Clinical Results of ART

The field of ART, as described in this review, has demonstrated great potential in enhancing clinical results by potentially providing more precise treatment through its capacity to adapt to temporal variations in anatomy during the course of treatment. An analysis of the dosimetric and clinical results documented in treated individuals offers a clear demonstration of the potential of this technique. Here, we survey a few key anatomical sites amenable to ART that have been widely reported in the literature.

### 5.1. Cervical Cancer

Cervical cancer provides a therapeutic challenge due to both the potential for intra-treatment responses as well as significant organ mobility with bladder and rectal changes during treatment. Studies have been conducted to determine the most effective PTV margins, as well as to measure and establish margins for adaptive therapy [39,40]. A retrospective study was conducted to examine the motion of target contours in a cohort of nine patients utilizing a CBCT-based system. The study compared non-adaptive and adaptive techniques and measured the changes in target contours before and after treatment. The results showed approximate movements of 0.32 cm in the anterior–posterior direction, 0.12 cm laterally, and 1.67 cm superior–inferiorly [41]. Using the quantifications they obtained, they assessed a 0.5 cm consistent expansion from the clinical target volume (CTV) to the PTV in order to encompass 98% of the motion variability. A validation cohort including 12 patients who underwent adaptive therapy demonstrated satisfactory coverage of the margin, with no adverse clinical consequences observed [41].

Previous research has conducted simulations of ART using CBCT, demonstrating its practicality. In a study conducted by Branco et al. (2023), a total of 15 patients were assessed [37]. For each patient, five random fractions were examined in an emulation environment. The target contours were adjusted and approved by a treating physician. The treatments were then tracked for dosage accumulation and compared with non-adaptive therapies. Branco et al. (2023) found that adaptive fractions resulted in a significant reduction in bladder and rectum D50% by an average of 37% and 36%, respectively, indicating a possible clinical advantage [37]. A recent study conducted on cervical and rectal cancer patients using emulators examined 13 patients and nearly 150 adaptive fractions. The study found that the average clinical work time for delivering adaptive treatment for cervical cancer was approximately 24 min. The study also observed an average improvement of 9.2% in V95% coverage, which was statistically significant. Additionally, there were small but statistically significant decreases in the D2cc to the bladder, bowel, and rectum [42]. A study conducted on 200 simulated sessions using CBCT-based adaptive techniques found that there was a 6% increase in the minimum dose received by the CTV. Additionally, there were slight but constant improvements observed in the maximum dose received by the colon, bladder, and rectum (D2cc) [43].

Early clinical outcomes have also been documented. A comparative efficacy study was conducted on seven patients with a gynecologic main condition who were treated using CBCT-based adaptive therapy. Prior to delivery, a physician thoroughly examined and authorized all therapies, with the main objective being to enhance CTV coverage. Out of the 61 fractions that were tracked, 45 of them received adaptive therapy. The average time it took to give the therapy was 33 min. Surprisingly, in 41% of the fractions, no modification of the CTV (clinical target volume) was needed because of the propagation of previously drawn outlines. Guberina et al. 2023 found that non-adapted treatments resulted in a median CTV coverage loss of 11.4%, whereas adapted treatments only had a loss of 0.6%. No grade 3 toxicity events were observed during the study, which had a median follow-up period of 4 months [10].

### 5.2. Lung Cancer

The potential benefits of ART are being assessed for lung cancer, considering its respiratory motion and the possibility of tumor shrinking during treatment. A conceptual investigation, involving a cohort of 12 patients, assessed several adaptive therapy options using the analysis and re-evaluation of previously acquired CBCTs. A study by Dial et al. (2016) found that implementing a daily adaptive method resulted in a mean increase of 1 Gy to the PTV [44]. Additionally, there was observed improvement in the doses received by the esophagus and spinal cord. In addition, a follow-up investigation of a CBCT-based system revealed that automatically generated contours of the target volume were comparable to manually adjusted contours, with an average volume difference of 2.86%. Furthermore, regardless of the adaptive workflow, there was a dosimetric enhancement of approximately 3% in the minimum dose (Dmin) when using automatic or edited contours [45]. In a separate study involving a unit that utilizes protons, five patients who finished their treatment were retrospectively analyzed. Their obtained CBCTs were recontoured, and adaptive proton-based plans were created. The study found that there was only a 0.6% variation in doses to the adaptive CTVs, in contrast to a 9.7% variation in the non-adaptive treatments [46]. Although there were fluctuations in lung V20, it was observed that non-adaptive therapy resulted in greater increases in circumstances when adaptive therapy would have increased V20, particularly when daily calculations and re-accumulation were conducted.

Furthermore, the initial clinical results of systems based on MRI have also been documented. A total of 54 lung cancers, with 29 being primary tumors, were assessed for the application of MRI-guided stereotactic body radiation treatment (SMART). According to the RTOG consensus criteria, 57% of the tumors were classified as central. Various prescriptions for SBRT were employed, with 55 Gy in five fractions and 60 Gy in eight fractions being used in 95% of the patients. The technique of gating was employed in a majority of patients, and the final plan received approval from physicians. The median duration of a SMART session was either 60 or 49 min, depending on whether Co-60 or MR-LINAC administration was used. The adaptive plan was employed 91% of the time, resulting in at least 100 BED coverage of the PTV in 93% of the treated tumors. The overall survival rate for patients with initial lung tumors over a two-month period was 83%, while the local control rate for all treated tumors over a 12-month period was 96%. The incidence of pneumonitis was 12%, observed in a total of six patients, with five of them having had prior lung radiation. The report emphasized the efficacy of an adaptable strategy in a group perceived by the treating institution to have a greater likelihood of unfavorable results [47].

A preliminary MRI-based phase I trial including five patients with ultracentral malignancies has reported outcomes after administering a total radiation dose of 50 Gy in five fractions. Out of a total of 25 fractions, adaptive therapy was used for 10 of them. The median follow-up period was 14 months. The primary cause for adaptation, observed in 70% of instances, was to fulfill an OAR constraint. There were no severe grade 3 toxicities according to the CTCAE v4 criteria, and the local control rate was 100% at 6 months [48].

### 5.3. Prostate Cancer

The presence of daily fluctuations in bladder and rectum volumes is well documented. Consequently, adaptive radiation therapy for prostate cancer holds considerable promise in terms of prospective advantages. Past research examining the retrospective recontouring of the bladder and rectum on a daily basis and the implementation of adaptive plans has consistently shown that there are improvements in rectal and bladder doses, while still maintaining target coverage, compared to the non-adaptive plans that were delivered [13]. A recent study by Moazzezi et al. (2021) examined the possibility of using AI-generated auto-contours with user changes in a CBCT-based system [36]. The study was conducted in an emulation environment and found that this approach resulted in a dosimetric benefit, with an average increase of 2.9% in CTV D98.

The clinical deployment of adaptive therapy has primarily been achieved through the use of MRI-based devices. An analysis was conducted on a group of 140 patients treated at a single institution. A total of 700 treatment sessions were administered. The majority of patients received a dose of 36.25 Gy delivered in five segments, with a margin of 0.3 cm around the PTV. The mean treatment time was 45 min, with a range of 40 to 70 min. The patients reported good tolerance of the treatment delivery, as assessed by patient quality of life questionnaires [49]. A phase I trial conducted at a single institution utilized a comparable technique to examine the results of 22 patients with prostate cancer. These patients received a total radiation dose of 36.25 Gy administered in five portions, with a corresponding 0.3 cm enlargement of the PTV. After a median follow-up period of 7.9 months, there were no severe (grade 3+) incidents recorded. However, there was a 22% incidence of moderate (grade 2) urinary toxicity at 3 months, with just one case persisting beyond 6 months. Additionally, there was a 4.5% occurrence of moderate (grade 2) rectal toxicity, which did not continue beyond 3 months. The patients’ quality of life scores indicated a restoration of normal function in general three months after therapy, as supported by the study conducted by Leeman et al. in 2022 [50]. The most extensive study conducted so far is a prospective phase II trial with 101 participants. The study utilized MRI-based online adaptive delivery and prescribed a dose of 36.25 Gy in five fractions. The incidence of Acute Grade II toxicity, as determined by the CTCAE v4 criteria, was 20% for the genitourinary (GU) domains and 3% for the gastrointestinal (GI) domains at the conclusion of the SBRT course. The rates decreased to 4.0% and 1.0%, respectively, after 3 months of follow-up, and the patient-reported quality of life aligned with this on the PR25 questionnaire. Rectal hemorrhage was reported only once, without the need for significant intervention [51].

### 5.4. Bladder Cancer

Bladder cancer is another disease site highly amenable to ART, as the target shape and size are directly affected by the bladder filling. Despite best efforts with water intake instructions, there exist daily variations in each treatment. Conventionally, in lieu of partially accounting for the variations with a large PTV margin of 1.5 to 2 cm, a plan library approach has also been proposed and utilized [52,53,54]. In this approach, a library of plans with varying levels of bladder fillings is prepared from the simulation, and the most suitable plan is selected from the library for treatment in each session based on the best match with the bladder seen on the verification image. This is in a way a poor man’s ART. While somewhat effective, the plan library approach has some inherent limitations. It is resource-intensive, from performing multiple simulations and preparing multiple plans to performing plan selection and record keeping at individual treatment sessions. Inter-observer variability in plan selection has been reported, and there are sometimes no suitable plans [55,56,57,58,59]. Moreover, the plan library approach is an approximation and not a real plan adaptation. Therefore, daily online ART could provide a better and more seamless approach to optimizing each treatment based on the treatment-day anatomy.

Azzarouali et al. reported success using the CBCT-based ART system Ethos to treat bladder cancer with a focal simultaneously integrated boost [60]. The visualization of the target and OARs was satisfactory with fiducial markers, with a median on-couch time of 22 min despite needing manual corrections on target contours for two-thirds of the sessions. Similarly, Astrom et al. reported Ethos ART treatments with an average CBCT-to-treatment time of 14 min. A few randomized trials have been initiated to investigate the efficacy of online ART for bladder cancer [15]. With its superior soft-tissue image contrast and ability for continuous intrafractional imaging, MR-LINAC has also been used for bladder cancer ART [61].

As online ART sessions often take longer than conventional sessions, intrafractional deformations can occur, and in the case of bladder cancer, continued bladder expansions can occur. An Ethos study by Pottgen et al. investigated this effect on focal bladder cancer treated with 5–10 mm PTV margins and found that daily ART is unnecessary and selective application of ART for significant deformations is sufficient to maintain effective dose coverage across the course due to the effectiveness of dose fractionation on mitigating dosimetric consequences of smaller target deformations occurring over the target surface [62].

### 5.5. Pancreatic Cancer, Liver Cancer, and Abdominal Oligometastasis

ART has been utilized in the application of stereotactic body radiation (SBRT) for the treatment of pancreatic cancer, liver cancer, and abdominal oligometastasis. The SBRT approach, which is gaining popularity, seeks to administer high-dose radiation to the tumors while minimizing damage to the adjacent OAR tissue. Nevertheless, the regular changes in the shape, size, and position of adjacent vulnerable serial OARs such as the duodenum, stomach, and small bowel significantly hinder the administration of high ablative doses to the targeted areas affected by these diseases. Consequently, ART is beneficial, as seen in the illustration presented in Figure 4 for the pancreatic case. MR-based ART systems have been particularly effective due to the significant soft-tissue contrast observed in the images [63].

A study by Bohoudi et al. (2017) described an approach for adaptive planning in pancreatic cancer SBRT that utilizes MR imaging to produce and modify outlines of OARs and target areas [25]. This strategy also involves partitioning the OARs based on their distance from the targets for the purpose of optimizing the treatment plan. The method exhibited rapid convergence and consequently produced adapted plans that fulfilled dosimetric standards more quickly than the traditional approach of contouring and planning the entire OAR. This resulted in reduced time for both the contouring and plan optimization stages of the adaptive workflow. In a similar vein, another study utilized adaptive planning to address the impact of pancreatic SBRT on OARs and targets that were recontoured within a 3 cm distance from the margin of the PTV [64]. Out of the 38 fractions administered to the eight patients in their group, the adaptive plan was selected for 26 fractions. This resulted in an average increase of 10.8% in the mean PTV V95% and a 12.6% increase in the CTV V98%. The average net online ART time, excluding patient positioning and treatment delivery, was 23 min. A study by Henke et al. (2016) found that in a group of 20 patients with abdominal primary and oligometastatic tumors who had MR-based ART treatment, more than 75% of the cases had a total session duration per fraction of less than 80 min [23]. Out of a total of 97 fractions, the adaptive plan was utilized for 81 fractions. Among these, 61 fractions were due to violations of hard constraints on OARs, while 20 fractions were due to identified chances for increasing the dose to the PTV, primarily in situations involving the liver. Furthermore, the adjustment of the plan resulted in an enhanced coverage of the PTV over the course of 64 treatment sessions. It is worth noting that no acute toxicities of grade 3 or higher were observed in the group under study. A recent study conducted at a single institution investigated the use of MR-based adaptive SBRT in 106 patients receiving abdominal or pelvic treatments. The study found that adaptive frequency was reduced, accounting for just 14% of all treatment sessions. Additionally, the study indicated positive outcomes in terms of local control and progression-free survival, with limited occurrence of toxicities [65].

While MR-based OAR avoidance ART has traditionally been the primary platform for abdominal ART treatments, alternative methods have also been utilized [34,66,67,68]. Several studies have demonstrated that the CBCT-based system is capable of producing high-quality images for the delineation of OARs and targets, and the use of ART leads to improvements in dose distribution [34,68]. The study by Ogawa et al. (2022) also mentioned that respiratory motion management techniques, such as end-of-exhalation breath hold, were found to be effective [68].

### 5.6. Head and Neck Cancer

Due to tumor shrinkage and patient weight loss during the course of treatment, head and neck radiotherapy also often leverages plan adaptation. Because the changes are usually more gradual, offline ART is often used with frequencies less than daily adaptation [69], although there are also ongoing trials investigating the applicability of daily online ART in head and neck cancer.

As with other cancer sites, ART can be beneficial in various ways depending on how it is used. One major way is to enable target margin reductions for reducing toxicities or further enabling dose escalation. For head and neck cancer, studies are underway evaluating PTV margin reductions down to as tight as 1 mm with ART [9,10]. The CBCT-based online ART system is highly amenable to such applications due to its seamless workflow and efficiency. Using Ethos, Dohopolski et al. simulated ART on oropharynx cancer patients with a 1 mm PTV margin which showed dosimetric superiority compared with conventional radiotherapy with a 5 mm margin, reducing on average 11 Gy for ipsilateral and 12 Gy for contralateral submandibular gland, 8 Gy for parotids, and similar levels of sparing for other relevant OARs, over a course with a 70 Gy highest prescription level [9]. Their phase II randomized trial of daily ART is underway to investigate patient-reported outcomes. Similarly, Guberina et al. investigated online ART on head and neck patients using Ethos and reported that while PTV margins of 5 mm or above may be sufficient for IGRT, further PTV margin reductions are feasible with ART to spare doses to the OAR [10].

## 6. Challenges and Outlook

ART is a groundbreaking advancement in cancer treatment, providing unmatched accuracy in personalized radiotherapy. This novel technique adapts in each session to the constantly changing characteristics of a patient’s anatomy and the changing size of the tumor during the course of treatment. Nevertheless, the process of incorporating ART into clinical practice is associated with complex obstacles and a continual requirement for improvement and advancement.

Despite integrating the various steps into one platform and automating most of the steps, the current online ART systems still contain some technical deficiencies. For example, the accuracy and robustness of OAR and target contouring still largely require careful inspections and frequent manual modifications by expert users. The treatment strategies are largely limited to IMRT due to the long dose computation time for VMAT. Currently, an ART session takes about 30–60 min for MRI-based systems and 15–30 min for CBCT-based systems. In addition to machine throughput and resource intensity for those professionals attending each session, this long session time increases patient discomfort and reduces patient compatibility and, more importantly, could lead to a wash-out of the dosimetric benefits gained by ART due to intrafractional organ deformations and movements.

There is a strong need for extensive clinical trials to determine the unique advantages of ART in different cancer sites. These investigations are essential for establishing the optimal frequency and procedures for ART, according to the specific characteristics of each disease. In addition, the present ART process, although markedly improved by AI compared with conventional radiotherapy workflow, is nevertheless burdened by time-consuming and labor-intensive processes that add to the time of each treatment fraction. Such added time has the potential to diminish the advantages of ART, especially when taking into account the changes in anatomical structure that occur throughout treatment and the challenges associated with patient suitability and comfort [70].

Continuing technological breakthroughs are thus necessary to improve the accuracy, robustness, and efficiency of AI-enhanced ART workflows. Efforts are also needed in addressing intrafractional anatomical changes and better incorporating advanced functional imaging methods. Furthermore, the substantial amount of data produced by ART requires more thought-out management and analysis approaches. The appropriate utilization of these data for applications such as radiomics and dosiomics is crucial for facilitating biological adaptation and customization of treatment regimens.

On the practical side, there needs to be a strong focus on the training, credentialing, and ongoing education of the clinical team participating in ART, encompassing radiation oncologists, medical physicists, dosimetrists, and radiation therapists. The main objective is to have a good grasp of the capabilities as well as limitations of the AI technologies used in the specific ART applications. Also, as ART introduces a new framework for radiation treatment, it necessitates a re-evaluation of roles, responsibilities, and the medical reimbursement models. Both offline and online ART can be resource-intensive. For online ART, currently a radiation oncologist and a medical physicist usually attend each session to review contours, make adjustments, select plans, and review the QAs and images. However, this level of resource intensity is very high and potentially prohibits the wide adoption and frequent application of online ART. To address this issue, the roles and responsibilities of different clinical parties as well as the medical reimbursement models need to be re-evaluated. Some have proposed new training and credentialing pathways for adaptive therapists or adaptive radiographers to take on the key role of daily ART decision-making and delivery [71]. Could a daily check or second review by the radiation oncologist and medical physicist be sufficient? Or even once weekly? Furthermore, the importance of the cost-effectiveness and accessibility of ART is a crucial matter that requires continuous focus. This also entails considerations of ethical factors associated with the development and deployment of medical AI technologies in ART, guaranteeing fair and unbiased availability and preventing inequalities in cancer treatment.

Continuing advancements in functional, biological, and molecular imaging as well as in AI would undoubtedly propel the further advancement of ART. Some recent evidence has emphasized the importance of the tumor microenvironment in the tumor’s response to radiation and underscored the significance of hypoxia in tumor radioresistance [72]. ART using PET scans with hypoxia-sensitive radiotracers such as fluoromisonidazole (FMISO), fluoroazomycin arabinoside (FAZA), and pentafluorinated etanidazole (EF5) can therefore enhance treatment response [73]. Recently, a trial protocol was published about the integration of weekly offline MRIs into the CBCT-guided ART of cervical cancer patients with Ethos, allowing additional functional assessment of tissue perfusion, hypoxia, or cellular density [14]. As time goes on, there might not be a clear line between different ART imaging modalities, and different imaging and treatment modalities could be combined to be more resource-effective.

Although ART shows great potential in transforming the results of radiation cancer treatment, it faces a range of challenges that encompass clinical, technological, practical, and ethical aspects. Tackling these difficulties requires a collaborative endeavor involving ongoing study, technical progress, creative thinking, and professional society recommendations and regulatory guidance in the domain.

## 7. Conclusions

ART represents a transformative approach in the field of radiation oncology, harnessing the power of advanced imaging and AI to adapt to the dynamic nature of patient anatomy during cancer treatment. By continuously adapting treatment plans to account for anatomical changes, ART significantly enhances the precision of radiation delivery, optimizes therapeutic outcomes, and minimizes damage to healthy tissues. This review discusses the motivation and workflow for ART and provides a comprehensive overview of the technical and clinical aspects of current ART systems and applications. As technology evolves, the integration of more sophisticated imaging methods and AI algorithms will further refine ART’s efficacy, ushering in a new era of personalized cancer treatment. The ongoing research and clinical trials in this domain are likely to expand the applicability of ART, promising improved care for cancer patients worldwide.

## Figures and Tables

**Figure 1 cancers-16-01206-f001:**
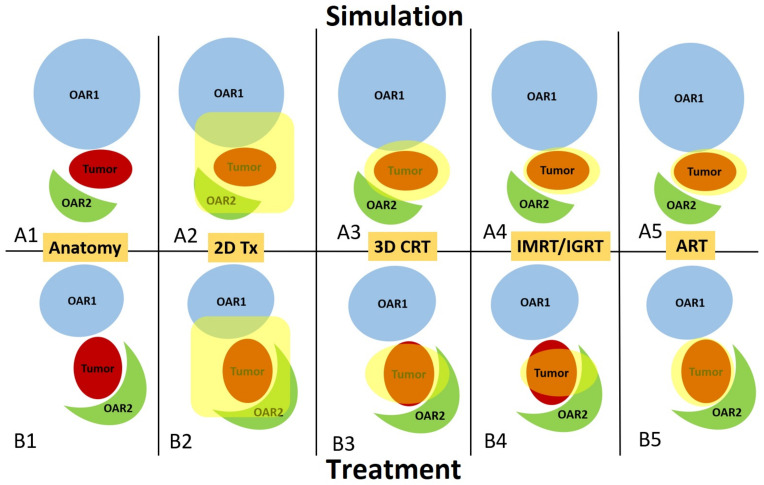
Illustrated here is a schematic comparing the anatomical changes and consequences of different radiation procedures between a simulation day (**A1**–**A5**) and a treatment day (**B1**–**B5**). Panels **A1** and **B1** depict the changes in shape and movement of the tumor (shown in red) and two organs at risk (represented in blue and green). This could be similar to a case of prostate cancer or cervical cancer (tumor), with bladder (OAR1) and rectum (OAR2) OARs. Panels **A2**–**A5** and **B2**–**B5** depict the volume that has been treated (indicated in yellow) on both the simulated anatomy and the anatomy on the day of therapy for the 2D treatment (2), 3D conformal (3), IMRT with IGRT (4), and ART (5) techniques, respectively.

**Figure 2 cancers-16-01206-f002:**
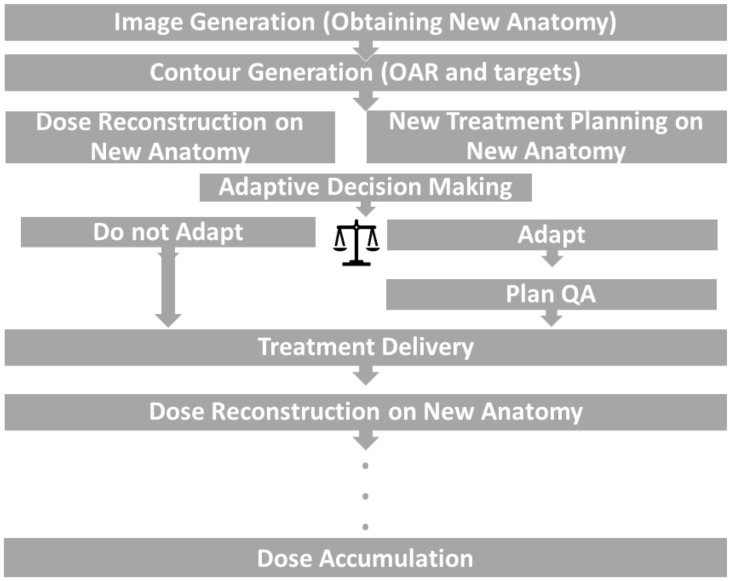
General workflow of ART.

**Figure 3 cancers-16-01206-f003:**
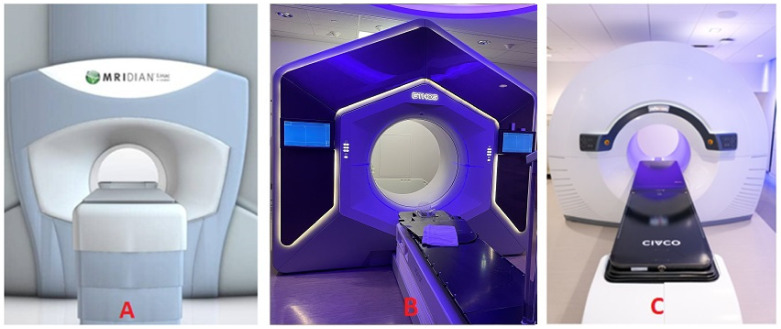
Example MR-based (**A**), CBCT-based (**B**), and PET-based (**C**) online ART systems, featuring the integration of corresponding imaging system with a 6MV FFF LINAC, installed at University of California Los Angeles, University of Rochester, and University of Texas Southwestern Medical Center, respectively.

**Figure 5 cancers-16-01206-f005:**
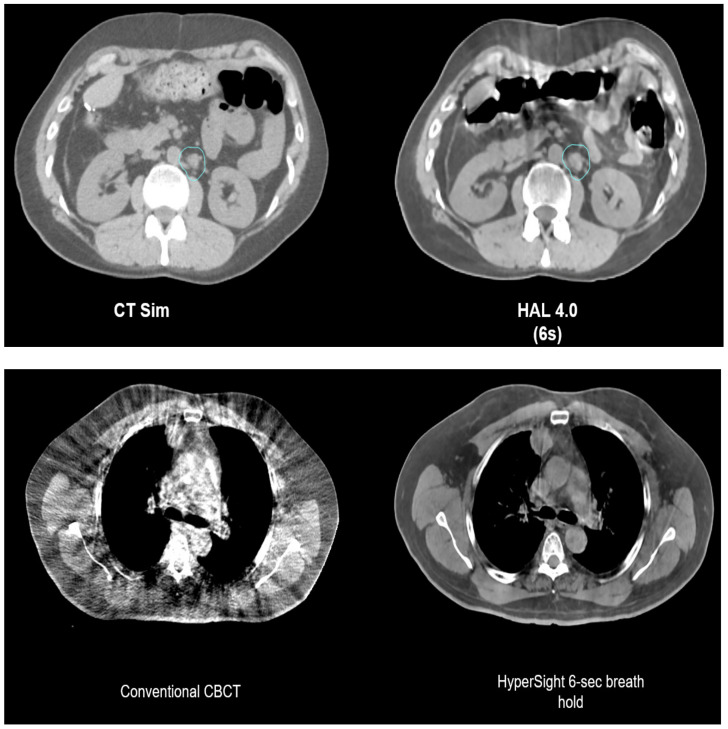
HyperSight CBCTs. The upper panel compares a simulation CT vs. a HyperSight CBCT (HAL 4.0 with a 6 s acquisition) of the abdomen. HyperSight CBCT shows comparable image contrast as the simulation CT and minimal streaking artifacts from gas pockets and breathing motion. The lower panel compares a conventional CBCT vs. a HyperSight CBCT (breath hold with a 6 s acquisition) of the thorax. The HyperSight CBCT shows much better image contrast and mitigated streaking artifacts and noise. Used with permission from Varian Medical Systems (https://medicalaffairs.varian.com/hypersight, accessed on 31 January 2024).

**Figure 6 cancers-16-01206-f006:**
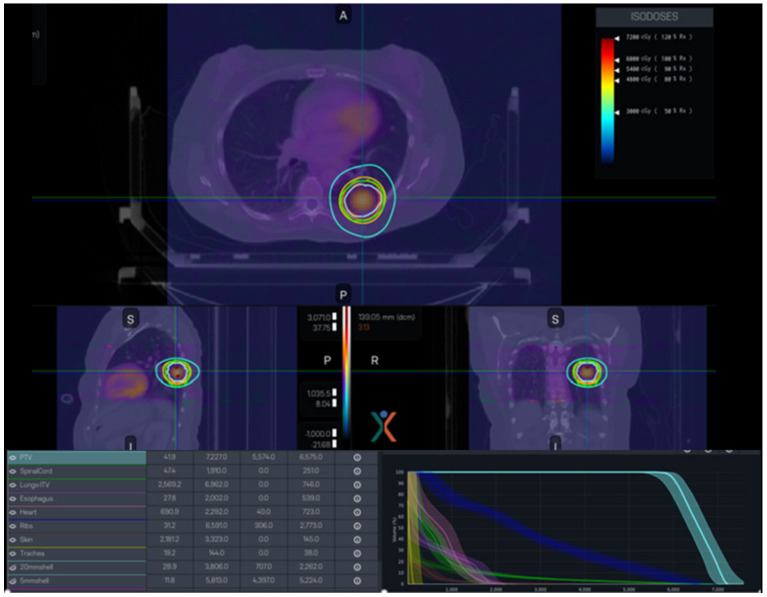
A representative SCINTIX radiation plan specifically designed for the treatment of lung conditions.

**Table 1 cancers-16-01206-t001:** Summary of offline vs. online ART.

	Offline ART	Online ART
Frequency	Offline ART involves evaluation/adjustments to the treatment plan in between treatment sessions, with the patient off the table. Plan adjustments are based on anatomy imaged at a certain timepoint and applied for later sessions. It is often applied in lower frequency such as mid-treatment, biweekly, or weekly.	Online ART involves evaluations/adjustments based on the session anatomy, while the patient stays on the treatment table, and is applied for the treatment of the same session. It is currently more often applied in each treatment session.
Complexity	When performed less frequently, it is generally less resource-intensive compared to online ART. At the same time, it could still be staff-time-demanding if offline ART has a less streamlined or automated workflow than available in online ART.	Online ART can be more complex and resource-intensive compared to offline ART because it requires specialized equipment and software and may be carried out more frequently.
Treatment planning	Offline ART is not conducted on patient images obtained in the session the adaptive plan is intended to be applied. Instead, planning is conducted offline on previously obtained images to apply in future sessions.	It allows for a highly individualized and precise treatment plan for each session, taking into account the new anatomy in each treatment session. The adaptive plan is made based on the session image and applied to the same session.
Clinical Applications	It is suitable for patients with tumors, OARs, and body habitus that are less likely to experience rapid anatomical changes and when the tumor is relatively distant from critical structures. It is commonly employed in situations such as head and neck cancers. Patient setup changes could also trigger the need for offline adaptation.	Used for cases where anatomical changes are expected on a daily basis. It is commonly employed in situations such as abdominal and pelvic malignancies. Based on the optimal trade-off between clinical benefits and required resources, the online ART platform may also be used for various disease sites to apply daily, weekly, or on-demand plan adaptation.

**Table 2 cancers-16-01206-t002:** A brief comparison of the three current online ART technologies.

	MRI	CBCT	PET
Current systems	Elekta Unity 1.5 T MRI with a 7MV FFF LINAC ViewRay MRIdian (legacy system) 6MV FFF 0.35 T MRI	Varian Ethos 6MV FFF	RefleXion X1 6MV FFF
ART workflow	Unity: Adapt to position (ATP) and adapt to shape (ATS). MRIdian: Choice between scheduled vs. adaptive plans.	Choice between scheduled vs. adaptive plans.	Offline ART feasible; online ART under development.
Strengths	Superior soft-tissue contrast; No radiation dose, can therefore provide continuous real-time monitoring during treatment; Functional and metabolic imaging.	Faster imaging speed, especially with HyperSight^TM^; High throughput; Cheaper and more accessible than the other two modalities; Planning directly on CBCT with HyperSight^TM^.	Metabolic and functional imaging; On-board kVCT provides good imaging quality; Real-time tracking; Multi-target delivery.
Limitations	Expensive; Longer imaging time and slow throughput; Need electron density for planning (MRIdian: DIR; Unity: bulk electron density); MR safety compatibility; Technical interference with LINAC.	Limited soft-tissue contrast; Radiation dose; No functional imaging.	Expensive; Longer imaging time; Radiation dose; Requires management of radiotracers; Need to combine with CT for anatomy and planning; Technical interference with LINAC.
Key clinical sites	Abdominal; Pelvic.	Pelvic; Head and neck; Breast; Lung.	Lung; Bone.

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
