# Peer review of "Adaptive Radiotherapy: Next-Generation Radiotherapy"

_cancers, 2024, doi:10.3390/cancers16061206_

Round 1
Reviewer 1 Report
Comments and Suggestions for Authors
this is a comprehensive plain language review, that gives a good overview about online ART. The only recommendation of this reviewer is to include an additional chapter on bladder cancer, as this entity can highlight the importance of the time factor underlying online adaptive radiotherapy and the effectiveness of dose fractionation on mitigating dosimetric consequences of smaller target deformations occurring over the target surface.
Author Response
Thank you very much for taking time to review our article. We appreciate your positive comments and insightful suggestion about bladder cancer. We have now added a section on bladder cancer (5.4) and discussed the important points you raised.
Reviewer 2 Report
Comments and Suggestions for Authors
This is a very good narrative review. There are just a few topics, that are important and have not been (sufficiently) addressed in my opinion.
The head and neck section is relatively brief. Studies are under way evaluating a very tight PTV margin of “down to” 1mm. Using this setting as an example, it should be pointed out clearly, that not the adaptation process itself is beneficial for the patient, but the reduction of PTV margins from the outset as well (which is enabled through sophisticated daily imaging). If one does not “give credit to” the PTV margin reduction from the outset, the benefit of oART might be underestimated if one only finds “minor dosimetric benefits” of the adapted plan compared to the scheduled plan.
Another topic, which might be of interest, is a thorough description and discussion of the “resource intensity” of online adaptive radiotherapy. Do we need a physician plus physicist always present at the online adaptive radiotherapy device? If not, is a daily check or “second look” by them enough, or even once weekly? Do we need to internationally propose the creation of a profession or further training in terms o a “oART radiographer”?
Lastly, just recently a trial protocol was published about the integration of weekly MRIs into the CBCT-guided radiotherapy of cervical cancer patients at the ETHOS: There might be no clear line between MR-guided/CT-guided/ and PET-guided radiotherapy and different imaging and treatment modalities could be combined to be more “resource effective”.
Author Response
Thank you for taking time to review our paper and providing valuable comments!
The head and neck section is relatively brief. Studies are under way evaluating a very tight PTV margin of “down to” 1mm. Using this setting as an example, it should be pointed out clearly, that not the adaptation process itself is beneficial for the patient, but the reduction of PTV margins from the outset as well (which is enabled through sophisticated daily imaging). If one does not “give credit to” the PTV margin reduction from the outset, the benefit of oART might be underestimated if one only finds “minor dosimetric benefits” of the adapted plan compared to the scheduled plan.
Thank you for the insightful suggestion. We have expanded the head and neck section and added relevant descriptions and discussion.
Another topic, which might be of interest, is a thorough description and discussion of the “resource intensity” of online adaptive radiotherapy. Do we need a physician plus physicist always present at the online adaptive radiotherapy device? If not, is a daily check or “second look” by them enough, or even once weekly? Do we need to internationally propose the creation of a profession or further training in terms o a “oART radiographer”?
Again, thank you for the insightful suggestion. In the "Challenges and outlook" section, we had some brief discussion on resource intensity and role/responsibility re-evaluation and re-definition. We have expanded these discussions in light of your comment, adding these detailed discussions.
Lastly, just recently a trial protocol was published about the integration of weekly MRIs into the CBCT-guided radiotherapy of cervical cancer patients at the ETHOS: There might be no clear line between MR-guided/CT-guided/ and PET-guided radiotherapy and different imaging and treatment modalities could be combined to be more “resource effective”.
Thank you for the helpful information. We have revised accordingly to expand the content and discussion in the last section.
Reviewer 3 Report
Comments and Suggestions for Authors
I would like to thank you the Authors for the interesting review.
The review analysed the role of ART
Major:
This is a narrative review not a systematic review
To long
Missing references in paragraph 2-3-4
In paragraph 5 explain the choice for the subsite
Minor comment:
Figure 1
I would suggest to sign on top the technique (only in the legend is a little bit difficult)
Author Response
Thank you for taking the time to review our manuscript and providing valuable comments!
We've added "narrative" review in our abstract.
We aimed to cover relevant topics and provide sufficient details on the topic. As you can see, other reviewers have requested additional content and subtopics which we've added in the revision. So, unfortunately, we are not in a position to shorten it.
Additional references have been added to parts 2-4.
Subsites that are most widely reported on with the use of ART are picked for Part 5. This explanation has been added in the leading paragraph of Part 5.
Figure 1 has been updated according to your suggestion.
Thank you again for your help in improving our review!
Reviewer 4 Report
Comments and Suggestions for Authors
Summary: A comprehensive narrative review on the existing knowledge of the application of adaptive radiotherapy (ART).
Comments:
1. Introduction-1st paragraph: "The goal is ... to kill tumors." It is suggested to complete the sentence as follows to convey the existing evidence on how radiotherapy really works: "The goal is ... to kill cancer cells or impede their proliferation while ...."
2. Introduction-2nd paragraph: "... then designing the customized beam arrangement, geometry, intensity, and modulation during plan optimization." Kindly add "beam number" to the list.
3. Kindly explain the acronyms in parentheses at their first appearance in the manuscript. Some acronyms have been explained more than once, for example, ART.
4. Figure 1 numbering. It is suggested to change 2A, 2B and so on into A2, B2, and so on. This will prevent confusion with subsequent Figures 2-6.
5. Figure 1. It is unclear whether the proposed model represents a real condition. It is suggested to provide an example reflecting a real anatomical position. The presented figure appears similar to the position of prostate cancer or cervical cancer (tumor) between the bladder (OAR1) and rectum (OAR2). If so, kindly label the components with the cancer type (prostate or cervical cancer) and the names of the OARs (OAR1 => bladder) and (OAR2 => rectum).
6. Figure 2 requires revision to enhance comprehensibility. The authors can simply use a (yes/no) sign to indicate the presence of a condition. This can improve the figure's readability.
7. Figure 5 requires additional explanation in either the main text or the figure caption. At first glance, in the first row, CT sim could provide a higher quality image with fewer artifacts compared to HAL 4.0. Additionally, kindly include the full term of HAL 4.0 in the figure caption.
8. Recent evidence has emphasized the importance of the tumor microenvironment in the tumor's response to radiation (https://pubmed.ncbi.nlm.nih.gov/38032584/). The referenced article underscores the significance of hypoxia in tumor radioresistance. Adaptive radiation using PET scans with hypoxia-sensitive radiotracers (e.g., FMISO, FAZA, EF5) can enhance treatment response (https://www.mdpi.com/2075-4426/12/8/1245). It is recommended that the authors mention this information in the "outlook" section, citing the two references provided.
9. It is expected that the authors provide an explanation of the challenges of ART at the end of the "Challenges and outlook" section.
Comments on the Quality of English LanguageAcceptable
Author Response
Thank you very much for taking the time to review our paper and providing valuable, detailed, and kind comments!
- Introduction-1st paragraph: "The goal is ... to kill tumors." It is suggested to complete the sentence as follows to convey the existing evidence on how radiotherapy really works: "The goal is ... to kill cancer cells or impede their proliferation while ...."
Revised accordingly. Thank you.
2. Introduction-2nd paragraph: "... then designing the customized beam arrangement, geometry, intensity, and modulation during plan optimization." Kindly add "beam number" to the list.
Revised accordingly. Thank you.
3. Kindly explain the acronyms in parentheses at their first appearance in the manuscript. Some acronyms have been explained more than once, for example, ART.
Checked again and corrected. Acronyms are now only explained at the first appearance in each independent part (simple summary, abstract, main text). Thank you.
4. Figure 1 numbering. It is suggested to change 2A, 2B and so on into A2, B2, and so on. This will prevent confusion with subsequent Figures 2-6.
Revised accordingly. Thank you.
5. Figure 1. It is unclear whether the proposed model represents a real condition. It is suggested to provide an example reflecting a real anatomical position. The presented figure appears similar to the position of prostate cancer or cervical cancer (tumor) between the bladder (OAR1) and rectum (OAR2). If so, kindly label the components with the cancer type (prostate or cervical cancer) and the names of the OARs (OAR1 => bladder) and (OAR2 => rectum).
It is a schematic depicting a fictitious condition but as you correctly stated, it shares similarities with a prostate or cervical case. We are concerned about labeling the schematic as representing a real condition for there could be objections the other way around. So we added in the figure caption "This could be similar to a case of prostate cancer or cervical cancer (tumor), with bladder (OAR1) and rectum (OAR2) OARs. ". Thank you for the valid comment and please kindly accept the compromised way to present this.
6. Figure 2 requires revision to enhance comprehensibility. The authors can simply use a (yes/no) sign to indicate the presence of a condition. This can improve the figure's readability.
Thank you for the comment. We've revised Figure 2 to enhance comprehensibility.
7. Figure 5 requires additional explanation in either the main text or the figure caption. At first glance, in the first row, CT sim could provide a higher quality image with fewer artifacts compared to HAL 4.0. Additionally, kindly include the full term of HAL 4.0 in the figure caption.
Thank you for the suggestion. Revised accordingly with added description in the caption.
8. Recent evidence has emphasized the importance of the tumor microenvironment in the tumor's response to radiation (https://pubmed.ncbi.nlm.nih.gov/38032584/). The referenced article underscores the significance of hypoxia in tumor radioresistance. Adaptive radiation using PET scans with hypoxia-sensitive radiotracers (e.g., FMISO, FAZA, EF5) can enhance treatment response (https://www.mdpi.com/2075-4426/12/8/1245). It is recommended that the authors mention this information in the "outlook" section, citing the two references provided.
Thank you for the insightful comment and helpful information. We've added discussion accordingly in the outlook section.
9. It is expected that the authors provide an explanation of the challenges of ART at the end of the "Challenges and outlook" section.
Thank you for the suggestion, In this section, we have briefly discussed the clinical, technological, practical, and ethical challenges of ART. In light of your suggestion, we further expanded these discussions, added a paragraph elaborating on the technical limitations of the current ART systems, and another paragraph discussing the resource-intensity requirements of ART.
Reviewer 5 Report
Comments and Suggestions for Authors
Dear authors,
It was a pleasure to have an opportunity to review this interesting rievew.
In this review, the authors provide a technical description of adaptive radiotherapy, an introduction to commercially available systems, and a detailed discussion of organ-specific indications. Adaptive radiotherapy is an evolving technology and this novel and comprehensive review will be of interest to the readers of Cancers.
The review is of good quality and there is nothing to comment on its content.
A minor comment is that the word "ART" appears for the first time in the text (except in the abstract) in the introduction (p2, l63), however it is not spelled out. Also, a footnote would be needed for "HAL 4.0" in Figure 5.
Regards,
Author Response
Thank you very much for your time reviewing our manuscript and for your positive comments.
Also thank you for the detailed review comments. We have revised both places according to your suggestions.
Round 2
Reviewer 3 Report
Comments and Suggestions for Authors
Any further comments